# ColCLIP: Enhancing Fine-Grained Image Retrieval with Pre-trained Embeddings

## Abstract

In the realm of image retrieval systems, efficiently searching for images based on any visual element described in the query is critical for user experience. However, current embedding models like CLIP primarily focus on aligning text with the most salient aspects of images, which may not always correspond to the elements users seek. In this paper, we propose ColCLIP, a fine-grained image retrieval system that leverages pre-trained embeddings and enhances them for our use case. We fine-tune CLIP on the Visual Genome Dataset and incorporate the MaxSim operator for image-text interaction. Our evaluations show that ColCLIP consistently outperforms standard CLIP in handling fine-grained retrieval tasks. ColCLIP improves image retrieval systems by enabling more relevant searches for users, while maintaining efficiency and ease of development. We release our code at `https://anonymous.4open.science/r/image-is-context-32B6`.

## 1 Introduction

In the ever-evolving digital landscape, the need for effective image retrieval systems is increasingly pertinent. Consider a scenario where an individual seeks to locate an image they have encountered previously. The most intuitive way to accomplish this is to input a text query describing the image, and the system then returns relevant images matching the query. However, an image inherently contains a plethora of information, with diverse objects and scenes. Individuals tend to focus on and recollect only specific elements of an image, making it a challenging task to formulate a query that captures the entirety of the image. For instance, in an image from the Visual Genome dataset Karpathy & Fei-Fei (2015) shown in Figure 1, queries like "a leaning pine tree", "a white car", or "a metal light post" should all have high alignment score with this image.

In light of this, we formulate our primary task as fine-grained image retrieval: the construction of encodings for both images and text in such a way that there is a high similarity score between the pair if the text matches any visual element in the image. The focus here is on "any" visual element, meaning that the text does not need to describe the most salient or prominent part of the image. This formulation can be analogized to a passage retrieval task in textual data where an image is akin to a passage, and each query represents text describing an object or aspect within the image.

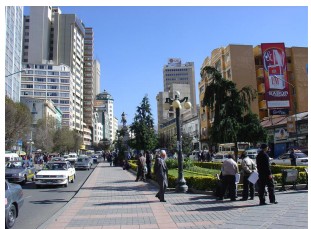

Figure 1: An example image from the Visual Genome.

One might argue that existing embedding models, such as CLIP presented in Radford et al. (2021), can be employed for this task. These models typically map images and text to a shared embedding space to compute similarities. However, while those models claim to have good zero-shot performance on a wide range of benchmarks, we posit that such models are not sufficiently equipped for our specific use case. This is primarily because these models are geared towards aligning text with a brief description of the entire image, usually focusing on the most salient aspect, which may not necessarily align with what the user remembers or queries for.

Nonetheless, there is an undeniable appeal in leveraging these pre-trained embedding models rather than building from scratch. For example, the training of the large Vision Transformer based CLIP model in Radford et al. (2021) involved 400 million image-text pairs and required 12 days on 256 V100 GPUs. Such computational resources are beyond the reach of most researchers. Additionally,

given that our fine-grained retrieval task is conceptually aligned with tasks where embedding models excel, their pre-trained weights can be invaluable.

In summary, we hypothesize that it is feasible to devise a fine-grained retrieval system that leverages the power of a pre-trained multimodal embedding model like CLIP, with minimal modifications. This adapted model is expected to significantly outperform the original model. This hypothesis holds great practical implications: it allows the development of a retrieval system that achieves both user experience and ease of development: users would find the system handy because they cause efficiently search for an image based on any object within it, while developers benefit from the feasibility and scalability offered by leveraging pre-trained models.

Building on our hypothesis, we present our core contributions in this paper. We introduce ColCLIP, a fine-grained image retrieval system, achieved by fine-tuning the CLIP model on the Visual Genome Dataset from Karpathy & Fei-Fei (2015) and incorporating the MaxSim operator from Khattab & Zaharia (2020) for efficient image-text interaction.

Through comprehensive evaluations involving three retrieval systems - CLIP, CLIP fine-tuned, and ColCLIP - we demonstrate that ColCLIP outperforms the other two. This observation not only supports our initial hypothesis but shows the effectiveness of the new architecture employed by ColCLIP. Furthermore, to provide insights into the behavior of our model, we conduct an analysis using saliency maps and integrated gradients. These analyses enable us to visually interpret how the model focuses on different regions in the image and different part of texts to understand how ColCLIP achieves enhanced performance for fine-grained retrieval tasks.

## 2    RELATED WORK

In the realm of constructing encodings for image and text that facilitate efficient retrieval based on the correspondence between text and any visual element in the image, several noteworthy approaches have emerged. Our work, in particular, is focused on achieving a balance between the expressiveness of image-text interaction and efficiency in retrieval.

The image-text alignment system presented by Karpathy & Fei-Fei (2015) is an early example that computes similarity scores between image captions and images, and establishes correspondence between words and image regions. Adapting this model to our task could be done through fine-tuning it on a dataset with region descriptions. However, its use of RNNs poses several challenges, such as difficulty in capturing contextual information efficiently and slow inference time due to the autoregressive property of RNNs. Moreover, the complexity of the architecture, which includes region proposals in R-CNN by Girshick et al. (2014) and explicit bounding box inference, can make the development process cumbersome. We may alternatively use this system to generate image captions as the index and match the query with generated captions using sentence transformers from Reimers & Gurevych (2019). However, this conversion leads to information loss as the image information is more comprehensive than the caption information Radford et al. (2021).

Recent advancements have led to models that project images and texts into a shared embedding space to gauge their similarities, as seen in Radford et al. (2021), Girdhar et al. (2023), Jia et al. (2021), and Li et al. (2022). These models primarily employ vision transformers Dosovitskiy et al. (2020) and encoder-only transformers for encoding images and texts. The reason is that they only employ a single embedding vector to encode images, which may not capture the fine-grained details needed for aligning text to any visual element in the image.

Contrastingly, the MiniGPT-4 system proposed by Zhu et al. (2023) adopts a different approach by projecting image embeddings and incorporating them into the input text for a large language model. This model can be adapted to our task via reasonable prompts like "`[image embeddings]` Does the image contain this: `[text embeddings]`.". While this method is highly expressive and could potentially excel at our proposed task, its practicality is hampered by the costly image-text interaction and the unpredictable nature of text output, which may not necessarily be in the form of a numerical similarity value.

Our work is notably inspired by research that emphasizes passage retrieval through adept late interaction techniques. The ColBERT model proposed by Khattab & Zaharia (2020) are of particular relevance. Rather than restricting itself to pooled embeddings, ColBERT preserves the embeddings

of each individual token and employs a MaxSim operator for late interaction. In our methodology, we take cues from ColBERT's MaxSim operator to adapt it to the multimodal domain. We will delve into the specifics of this interaction mechanism in our methods section. Another work inspired by the ColBERT model is the FILIP model Yao et al. (2021). While both approaches incorporate rich embeddings and MaxSim for late interaction, we distinguish ourselves in several ways. FILIP's approach centers on training models from scratch and highlighting zero-shot capabilities, whereas our experiments shed light on the robustness of pretraining, even when employing weights from an alternate infrastructure. However, the most significant distinction lies in our downstream tasks. FILIP focuses on zero-shot performance, whereas our task of fine-grained image retrieval aligns images with finer level object captions instead of general image captions, introducing a significantly greater level of complexity. In this demanding context, our model, ColCLIP, not only surpasses the baseline CLIP but does so by a substantial margin, showcasing the power of rich embeddings when compared to single embeddings and ColCLIP's effectiveness in tackling a more intricate and challenging downstream task. In addition, we offer publicly accessible source code and plan to release model weights as a commitment to reproducibility.

## 3   ColCLIP Method

### 3.1   Model Architecture and Interaction

Our proposed model, ColCLIP, is an extension of CLIP from Radford et al. (2021), wherein we introduce an alternative method for interaction between image and text embeddings. It is essential to mention that the architecture of both the text encoder and the image encoder in ColCLIP is identical to that in CLIP. This design choice enables us to efficiently leverage the pre-trained weights.

The image encoder in ColCLIP adopts the Vision Transformer from Krishna et al. (2016).

Initially, the image is center cropped, normalized, and resized to $224 \times 224$. Then, a CNN layer processes the image, converting it into a series of patches. These patches are then fed into a transformer, resulting in the final hidden states. Concurrently, the text encoder, an encoder-only transformer, processes the textual input. Both encoders use a linear projection layer that maps the final hidden states into a common embedding space, facilitating interaction between the modalities.

Let's represent the projected hidden state of the image encoder as $E_i \in \mathbf{R}^{T_i,d}$, and that of the text encoder as $E_q \in \mathbf{R}^{T_q,d}$. Here, $T_i$ denotes the number of patches in the image, while $T_q$ represents the number of tokens in the query. For simplicity, we assume that each embedding in $E_i$ and $E_q$ is normalized. In CLIP, the final encoding of images and text relies solely on the hidden state of a special token [EOT]. Mathematically, this is expressed as $S_{i,q} := E_i[t_i] \cdot E_q[t_q]$, where $t_i$ and $t_q$ correspond to the position of the special token within the sequence, as in Radford et al. (2021).

However, ColCLIP takes a different approach, inspired by Khattab & Zaharia (2020), and utilizes the entire hidden states through a MaxSim operator for computing similarity, defined as:

$$S_{i,q} := \frac{1}{T_q} \sum_{j \in [T_q]} \max_{k \in [T_i]} E_q[j] \cdot E_i[k]$$

This alternative interaction is motivated by the intuition that MaxSim, which employs the full hidden states, is more expressive to the pooled embedding approach in CLIP. By capturing more information, ColCLIP is poised to enhance the fine-grained retrieval capabilities, making it adept at handling a range of queries aligning to various visual elements in images. The similarity score is then scaled by a learnable scalar to be interpreted as logits. For retrieval task, we select images with highest similarity score with a given query.

In addition to the modification in interaction through the MaxSim operator, ColCLIP undergoes fine-tuning on a task-specific dataset. Through fine-tuning, ColCLIP is instructed to comprehend the nature of the task where the query does not necessarily need to be aligned with the most salient part of the image. Instead, it gains the ability to focus on various fine-grained details, regardless of their prominence within the image. This ensures that ColCLIP is not just more expressive due to the interaction method but is also adept at handling the diverse range of queries that users might pose.

## 3.2 ASSYMMETRIC TRAINING

ColCLIP has similar training pipeline as CLIP with slight modifications.

During the training phase, in each step, we select a batch consisting of $N_i = 64$ images. A notable deviation from the original CLIP training pipeline is that for each image in the batch, we sample $c = 5$ labeled captions, instead of just one ($c = 1$ in CLIP). This decision is driven by the observation that the queries associated with each image exhibit a broad range of diversity. The end result is a total of $cN_i$ image-text pairs to work with during a single training step.

Following the formation of these pairs, we employ an adapted version of the InfoNCE loss from van den Oord et al. (2019). For every query, we are presented with $N_i$ candidate images, and the objective is to maximize the alignment score of the correct image while concurrently minimizing the scores of incorrect images. It's worth highlighting that the original CLIP architecture incorporates a symmetric approach, wherein for each image, it seeks to maximize the score of the correct text and minimize the score of incorrect text. However, we specifically choose to drop this symmetry, as this aspect is not directly relevant to our image retrieval use case. We present the pseudocode in Appendix Section A

## 3.3 SPACE OPTIMIZATION USING DIMENSIONALITY REDUCTION

In real-world scenarios, the storage and computational requirements can pose significant challenges. As mentioned in Section 3.1, ColCLIP-Large produces image embeddings with a size of $768 \times \text{num\_patches}$, which can be quite large. To address this, we introduce a compressed variant of ColCLIP-Large, called ColCLIP-Large-lite, which reduces embedding dimension size from 768 to 128 while attempting to preserve the critical features required for efficient and accurate image retrieval. We employ a linear projection to accomplish this task, such that projected image embedding $E_i^{'}[j] = \mathbf{W}_i E_i[j]$ and projected query embedding $E_q^{'}[j] = \mathbf{W}_q E_q[j]$ where $\mathbf{W}_i$ and $\mathbf{W}_q$ are learnable projection matrices.

# 4 EXPERIMENTS

## 4.1 SETUP

**Dataset**  Our experiments utilize the Visual Genome dataset, introduced by Krishna et al. (2016), which comprises 108,077 images and 5.4 million region descriptions. Each entry contains an image ID, and a list of texts describing different objects and aspects of the image. Sourced from a combination of the YFCC100M (Thomee et al. (2016)) and MS-COCO (Lin et al. (2015)) datasets, these images represent a diverse range of content. The captions are crowd-sourced through Amazon Mechanical Turk, using an annotation process that emphasizes capturing a wide range of objects in each image. The captions' diversity and granularity make it a robust choice for fine-tuning and evaluating our model.

**Model**  In our experiments, the ColCLIP model is instantiated from two distinct architectures originating from CLIP, namely CLIP-base and CLIP-large. While we adopt their architecture and pretrained weights, we modify their interactions, resulting in two variants of our ColCLIP model: ColCLIP-base and ColCLIP-large.

**Query Sets**  To comprehensively evaluate retrieval systems, we utilized the test split of the Visual Genome dataset to create diverse evaluation query sets. These included simple query set with varying index sizes (100, 1000, 5000, and 10808) to assess performance across different scales. We employed a sentence transformer model [1] to identify image captions that are similar to the sampled query to include these images in the target label as well. In addition, we included the ImageNet validation dataset, consisting of 50,000 images and 1,000 classes, to assess the system's ability to capture the most salient part of an image (the main task of CLIP from Radford et al. (2021)), in addition to the fine-grained details measured in the previous evaluation query sets. The detailed procedure for constructing these query sets is elaborated in the Appendix Section B.

---

[1] https://huggingface.co/sentence-transformers/all-MiniLM-L6-v2

**Evaluation Metrics** We evaluate ColCLIP using a multi-faceted set of metrics: Success@$K$, Precision@$K$, Recall@$K$, and Average Precision, with $K$ set at 1, 5, 10, and 25. Success@$K$ gauges the general retrieval performance, while Precision@$K$ and Recall@$K$ offer nuanced insights into the system's accuracy and comprehensiveness, respectively. Average Precision provides a holistic evaluation sensitive to rank, precision, and recall. To account for system efficiency, we also measure query latency in milliseconds.

**Baselines** We compare ColCLIP's performance against two key baselines. The first, "CLIP Frozen," serves as a measure of CLIP's zero-shot capabilities in fine-grained image retrieval. The second, "CLIP Finetuned," involves fine-tuning the original CLIP model on the Visual Genome dataset, allowing us to assess the benefits of dataset-specific adaptation without altering the model architecture. These baselines enable us to isolate and evaluate the contributions of our architectural modifications, such as the MaxSim operator, to performance gains.

## 4.2 COMPARISON WITH BASELINES

We compare ColCLIP-Large with CLIP-Large on a simple evaluation query set with an index size of 1000. The evaluation metrics are summarized in Table 1. It is evident that the ColCLIP Large model dominates the CLIP-Large Frozen model. Furthermore, ColCLIP also outperforms the CLIP Large Finetuned model, suggesting that the high performance of ColCLIP is not solely attributed to finetuning, but also to the ColCLIP architecture itself, especially the MaxSim Operator that enables richer text-image interactions. In contrast to the CLIP model that uses a single pooled output to obtain a single embedding for each image and text, ColCLIP preserves all last hidden states and enables richer MaxSim interactions, resulting in higher performance. This improvement is not limited to a specific image index size, as observed in Table 7 in Appendix, where ColCLIP Large consistently outperforms CLIP Large Finetuned across different index sizes ranging from 100 to 10,808, with the only exception of recall@5 for an index size of 10,808.

| Model | AvgPrecision | Success@K | | | Precision@K | | | Recall@K | | |
|---|---|---|---|---|---|---|---|---|---|---|
| | | K=1 | K=5 | K=10 | K=1 | K=5 | K=10 | K=1 | K=5 | K=10 |
| CLIP Large Frozen | 0.223 | 0.141 | 0.308 | 0.405 | 0.150 | 0.068 | 0.046 | 0.141 | 0.306 | 0.404 |
| CLIP Large Fine-tuned | 0.364 | 0.234 | 0.499 | 0.634 | 0.242 | 0.110 | 0.072 | 0.228 | 0.501 | 0.638 |
| ColCLIP Large | **0.406** | **0.281** | **0.536** | **0.677** | **0.293** | **0.117** | **0.076** | **0.281** | **0.535** | **0.674** |

Table 1: Quantitative Metrics for Retrieval Models on Simple Query Set of 1000 Index Size

When evaluating runtime efficiency on Google Colab with an A100 GPU, we found that CLIP Large Finetuned has an average query latency of 12.291 ms/query. In comparison, ColCLIP Large clocks in at 14.497 ms/query. Despite the advanced features of ColCLIP, this modest increase of just 2.206 ms/query suggests that our model introduces only a negligible overhead, positioning COLCLIP as an efficient and practical choice for real-time applications.

## 4.3 PERFORMANCE ON NON-UNIFORM TEST SET

**Challenging Query** We created a challenging query set specifically designed to query objects occupying less than 5% of the image area.

**Compositional Query** We created compositional evaluation query sets by concatenating two sampled single queries in the same image.

The results of these evaluations are reported in Table 2. The result demonstrates that the ColCLIP Large model is not only proficient in retrieving uniformly sampled queries but also excels in handling challenging queries and compositional queries. In the case of the challenging query set, which

focuses on objects occupying less than 5% of the whole image, the model exhibits only a slight decrease in all metrics, indicating its ability to attend to fine-grained details as hypothesized. Additionally, when evaluating compositional queries, the average precision significantly increases from 0.406 in the uniform evaluation query set to an average precision of 0.621, showcasing the model's strength in retrieving compositional queries even without explicitly fine-tuning on compositional data. This aligns with our expectation that the MaxSim operator enables more comprehensive interactions between images and queries, allowing the retrieval system to assign higher scores to images that possess high similarity patches with every token embedding of the query.

| Query Set | AvgPrecision | Success@K | | | Precision@K | | | Recall@K | | |
|---|---|---|---|---|---|---|---|---|---|---|
| | | K=1 | K=5 | K=10 | K=1 | K=5 | K=10 | K=1 | K=5 | K=10 |
| Uniform (Full) | 0.406 | 0.281 | 0.536 | 0.677 | 0.293 | 0.117 | 0.076 | 0.281 | 0.535 | 0.674 |
| Challenging | 0.379 | 0.242 | 0.524 | 0.660 | 0.262 | 0.118 | 0.076 | 0.244 | 0.516 | 0.648 |
| Compositional | 0.621 | 0.500 | 0.775 | 0.867 | 0.500 | 0.155 | 0.087 | 0.500 | 0.775 | 0.867 |

Table 2: Metrics for ColCLIP-Large on Simple, Challenging, and Compositional Query Sets

## 4.4 IMPACT OF EMBEDDING SIZE ON PERFORMANCE

Table 3 displays a comparison of the foundation model sizes. We included both base and large models of CLIP finetune and ColCLIP for comparison purposes in this table.

| Model | AvgPrecision | Success@K | | | Precision@K | | | Recall@K | | |
|---|---|---|---|---|---|---|---|---|---|---|
| | | K=1 | K=5 | K=10 | K=1 | K=5 | K=10 | K=1 | K=5 | K=10 |
| CLIP Base Fine-tuned | 0.303 | 0.190 | 0.422 | 0.563 | 0.201 | 0.095 | 0.064 | 0.187 | 0.421 | 0.559 |
| ColCLIP Base | 0.291 | 0.173 | 0.418 | 0.549 | 0.184 | 0.093 | 0.062 | 0.169 | 0.417 | 0.541 |
| CLIP Large Fine-tuned | 0.364 | 0.234 | 0.499 | 0.634 | 0.242 | 0.110 | 0.072 | 0.228 | 0.501 | 0.638 |
| ColCLIP Large | **0.406** | **0.281** | **0.536** | **0.677** | **0.293** | **0.117** | **0.076** | **0.281** | **0.535** | **0.674** |

Table 3: Quantitative Metrics for Base and Large Retrieval Models on Simple Query Set of 1000 Index Size

The result reveals that ColCLIP-Large outperforms ColCLIP-Base, primarily because the larger model contains more parameters and is more expressive in capturing image-text relations. However, it is worth noting that the ColCLIP-Base model slightly underperforms compared to CLIP Base Finetuned. Unlike the performance gain observed between ColCLIP-Large and CLIP-Finetuned Large, the base model does not benefit from the architecture modification. This can be attributed to the base model being less expressive than the large model, where the interaction of CLIP, which computes similarity between pooled embeddings, may be sufficient for text-image interactions in this case.

To further investigate the performance divergence observed between the ColCLIP Base model and the CLIP model, we conducted subsequent experiments by training both models from scratch. These experiments aimed to elucidate whether the observed performance difference could be attributed to the use of incompatible pre-trained weights for ColCLIP. The results reveals that when both models are trained from scratch, there is no significant disparity in validation loss and evaluation results as shown in Table 4. This underscores the hypothesis that, when initiated without the imposition of incompatible pre-trained weights, both models attain similar performance levels.

Nonetheless, since the Large Model produces richer embeddings for individual text tokens and image patches, simply relying on the single pooled output may discard useful information. In this

| Model | AvgPrecision | Success@K | | | Precision@K | | | Recall@K | | |
|---|---|---|---|---|---|---|---|---|---|---|
| | | K=1 | K=5 | K=10 | K=1 | K=5 | K=10 | K=1 | K=5 | K=10 |
| CLIP Base | 0.054 | 0.023 | 0.079 | 0.126 | 0.025 | 0.019 | 0.015 | 0.021 | 0.078 | 0.127 |
| ColCLIP Base | 0.049 | 0.022 | 0.069 | 0.125 | 0.024 | 0.015 | 0.014 | 0.020 | 0.065 | 0.118 |

Table 4: Quantitative Metrics for Retrieval Models with uninitialized weights on Simple Query Set of 1000 Index Size

scenario, the ColCLIP architecture effectively preserves all meaningful embeddings and achieves a performance boost from the MaxSim operation.

## 4.5 COLCLIP-LARGE-LITE: REDUCE EMBEDDING SIZE WITHOUT SACRIFICING PERFORMANCE

Drawing from section 4.3's findings where the performance of COLCLIP was observed to diminish with smaller embeddings, we introduced additional optimization strategies to enhance the embedding's efficiency without sacrificing its effectiveness. The result of this optimization is ColCLIP-Large-lite.

| Model | AvgPrecision | Success@K | | | Precision@K | | | Recall@K | | |
|---|---|---|---|---|---|---|---|---|---|---|
| | | K=1 | K=5 | K=10 | K=1 | K=5 | K=10 | K=1 | K=5 | K=10 |
| ColCLIP Large | 0.406 | 0.281 | 0.536 | 0.677 | 0.293 | 0.117 | 0.076 | 0.281 | 0.535 | 0.674 |
| ColCLIP-Large-Lite | 0.393 | 0.265 | 0.511 | 0.667 | 0.279 | 0.113 | 0.076 | 0.265 | 0.511 | 0.665 |

Table 5: Quantitative Metrics for Original and Lite Retrieval Models on Simple Query Set of 1000 Index Size

Table 5 presents a performance comparison between the original ColCLIP and ColCLIP-large-lite. Impressively, ColCLIP-large-lite manages an average precision of 0.393, only marginally below the 0.406 recorded by the original ColCLIP. Despite this slight difference, key metrics such as precision@1, recall@1, and success@1 remain robust: for instance, precision@1 for ColCLIP-large-lite stands at 0.2790, compared to 0.2930 for ColCLIP. This becomes especially significant when considering the reduction in dimensionality—from 768×num_patches to 128×num_patches. The performance retention coupled with this reduced dimensionality results in substantial storage and computational savings, making ColCLIP-large-lite an outstanding choice for scenarios demanding both performance and efficiency.

## 4.6 EFFECT OF FINETUNING ON ZERO-SHOT ABILITIES

| Base Model | Accuracy | Large Model | Accuracy |
|---|---|---|---|
| CLIP Base Frozen | 0.5554 | CLIP Large Frozen | 0.6688 |
| CLIP Base Finetuned | 0.3351 | CLIP Large Finetuned | 0.4862 |
| ColCLIP Base | 0.2964 | ColCLIP Large | 0.4944 |

Table 6: Zero-shot Classification for Retrieval Models

Table 6 displays each retrieval model's zero-shot classification performance on the validation split of ImageNet dataset.

In contrast to the original CLIP Frozen Model, which showcases its zero-shot transfer ability in classification tasks, we observed a degradation in the model's zero-shot classification performance after finetuning. Table 6 illustrates that both CLIP Finetuned and ColCLIP experience a decrease

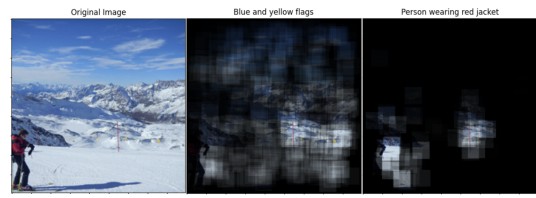

(a) Saliency Map Using CLIP Large Finetuned Model

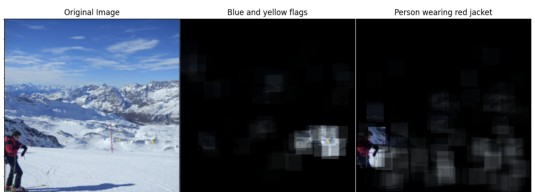

(b) Saliency Map Using ColCLIP Large Model

Figure 2: Saliency Map on queries "Blue and yellow flags" and "Person wearing red jacket"

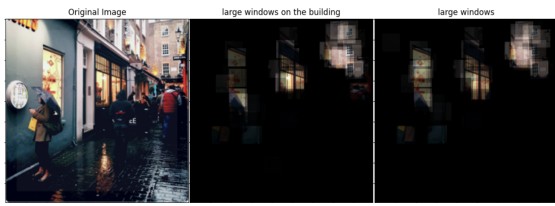

(a) Saliency Map Using CLIP Large Finetuned Model

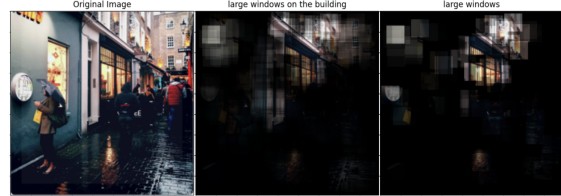

(b) Saliency Map Using ColCLIP Large Model

Figure 3: Saliency Map on queries "large windows on the building" and "large windows"

in classification accuracy compared to CLIP Frozen. This indicates that while the retrieval system achieves better performance in fine-grained information retrieval, its capability to capture general information from the entire image diminishes as a result of finetuning on the Visual Genome dataset.

## 4.7 QUALITATIVE ANALYSIS USING SALIENCY MAP

In order to gain insight into the regions of an input image that our model focuses on, we employ feature attribution saliency map, which highlights the particular sections of the image that the model deems significant in its computations.

We initiate the saliency map as a zero array having the same dimensions as the image. The alignment score between the image and the query is computed initially. Subsequently, we go through a series of iterations where a rectangular portion of the image is randomly masked out, and the alignment score is recomputed. If there is a decline in the score, the absolute difference in the score is added to the corresponding region of the saliency map. This iterative process is repeated multiple times, which allows for a more robust visualization. We also acknowledge the limitation of the current saliency map technique, which is explained more in Appendix E.

Figure 2 examines the effectiveness of CLIP Large Finetune Model and ColCLIP Large Model in attending to specific image regions using saliency maps generated through random-masking on the original image (first column), with input queries of "Blue and yellow flags" (second column) and "Person wearing red jacket" (third column). When queried about "Blue and yellow flags", CLIP's saliency map (Figure 2a) exhibits a broader focus on the overall image rather than pinpointing the exact region of interest. In contrast, ColCLIP's saliency map (Figure 2b) exhibits exceptional precision by attending to the specific area of the blue and yellow flags, which is even hard for humans to detect at first glance. Similarly, for the query "person wearing red jacket," CLIP's attention is divided between the skier wearing the red jacket and the red pole in the center of the image. This observation highlights the significant influence of the word "red" on image and text alignment in CLIP. In contrast, ColCLIP consistently prioritizes the actual skier, delivering accurate alignment with the expected region of interest. These findings underscore ColCLIP's outstanding performance in attending to detailed image areas.

While ColCLIP demonstrates remarkable attention to fine details in an image, it exhibits limitations when attending to more salient features. In Figure 3, we observe these differences in performance. When the query is "large windows on the building," CLIP (Figure 3a) focuses primarily on the tar-

geted region of the window on the building and parts of the building itself. In contrast, ColCLIP (Figure 3b) directs attention to a larger area encompassing both the building and the windows. However, when prompted with the query "large windows," both CLIP and ColCLIP attend to similar regions. CLIP identifies separate regions, including the large window in the front and the windows on the buildings behind, while ColCLIP primarily focuses on the large window within the image. This discrepancy highlights the inefficiency of ColCLIP when tasked with identifying the most salient feature in an image. While ColCLIP excels at capturing intricate details, it struggles to precisely isolate and emphasize the specific salient feature as requested by the query.

### 4.8 QUALITATIVE ANALYSIS WITH INTEGRATED GRADIENTS

To further investigate the relationship and interactions between text and images in ColCLIP, we leverage the integrated gradient feature attribution method Sundararajan et al. (2017) to thoroughly examine the contribution of each word to the final outcome. Specifically, we employ the Layer Integrated Gradient technique provided by Captum Kokhlikyan et al. (2020) to attribute importance scores to the inputs of ColCLIP's text projection layer. The details about the procedure of using the integrated gradient is shown in Appendix F.

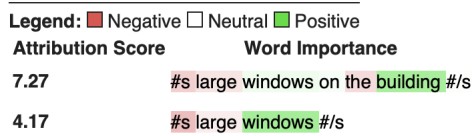

Figure 4: Integrated Gradients on Queries in Figure 3

In Figure 4, we applied integrated gradients to investigate the queries that ColCLIP struggled to identify in Figure 3. Words highlighted in green indicate the most influential features contributing to the target label, while words in red negatively contributes to the target label. Examining the resulting word importance, we can observe that the presence of the word "building" holds more significance, leading the model to primarily focus on the building itself rather than the intended "large windows" that were the target of the query. When we remove the confounding factor of "building" from the prompt and solely use the query "large windows," ColCLIP correctly identifies "windows" as the most important word, leading to the successful retrieval of images with similar attended regions as observed in CLIP. This outcome highlights the correlation between texts and images during the alignment process and the need for further enhancements in text projection to accurately identify objects and prioritize the most critical words in the input query.

## 5 CONCLUSION

In this work, we start with the hypothesis that it is feasible to build a fine-grained retrieval system by harnessing the potential of a pre-trained multimodal embedding model like CLIP, with minimal modifications, and that this adapted model, namely ColCLIP, would substantially outperform the original model in scenarios where the original is used in a zero-shot manner.

Firstly, we constructed ColCLIP by building upon CLIP, demonstrating its superior performance compared to the zero-shot capabilities of the original CLIP. This confirms the positive outcome of our hypothesis. Secondly, we conducted experiments to discern why ColCLIP surpasses the performance of the original and frozen CLIP model. We evaluated the relative contributions derived from fine-tuning as compared to those attributable to architectural enhancements. Thirdly, our study incorporated an in-depth examination to identify the types of queries where the original CLIP, when fine-tuned, excels and those better handled by ColCLIP. We employed saliency maps and integrated gradients for feature attribution, which aided in gaining insights into the model's behavior.

As a direction for future research, there is an opportunity to evolve our model into a multi-task system that not only excels in fine-grained image retrieval but also retains the impressive zero-shot performance of CLIP across a variety of tasks. Additionally, by incorporating various advancements from ColBERTv2, as introduced by Santhanam et al. (2021), there is potential to further reduce retrieval times, making ColCLIP even more efficient in practical applications.

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

## A   Pseudocode of Our Training Process

We present the pseudocodoe of one training step here. We also include the full training pipeline in our released code.

```
# extract image and query features
I_f = image_encoder(I) # [N_i, T_i, D_i]
T_f = text_encoder(T) # [cN_i, T_t, D_t]

# project to joint space
I_e = normalize(I_f @ W_i, axis=2)
T_e = normalize(T_f @ W_t, axis=2)

# get pairwise similarities [N_i, cN_i]
logits = interact(I_e, T_e)

# assymetric loss function
labels = [[i] * c] for i in range(N_i)]
loss = cross_entropy_loss(logits, labels, axis=0)
```

## B   Evaluation Query Set

In order to thoroughly evaluate the performance of our retrieval system and compare it with the baseline model, we created several evaluation query sets using the test split of the Visual Genome dataset. The test split consists of a total of 10,808 images along with their corresponding region captions. To assess the impact of index size on the system's performance, we generated query sets with different index sizes: 100, 1000, 5000, and 10808. These query sets have the same number of queries 1000.

To construct each evaluation query set, we uniformly sampled a number of images corresponding to the index size and extracted all captions associated with these images. From these captions, we randomly sampled 1000 captions to form our queries. The corresponding images were then treated as the targets for the retrieval task. During the sampling process, We observed that the multiple images may contain identical or highly similar captions with slight variations in phrasing. To address this, we employed a sentence transformer model [2] to generate embeddings for the captions. By computing the cosine similarity between all captions with sampled queries, we identified and included images with highly similar captions in the target label as well. Notable examples of similar captions we identified include: "a cat's black tail" and "black cat tail," "the wall is green" and "green color on the wall." Through this data augmentation process, our objective is to narrow the discrepancy between the gold label in our validation dataset and the actual ground truth for each query. This approach ensures the reliability of the validation data, enabling it to provide an accurate recall during evaluation. By doing so, our evaluation becomes a more precise estimation of the system's capacity to generalize and perform effectively in real-world scenarios. However, we acknowledge that this process is automated and there is a possibility of missing target images. In this case, our evaluation query set may underestimate the precision@k and success@k metrics of the system, as it's possible that some valid retrieved images are not included in our gold label. The impact on the recall@k metric remains undetermined. To enhance the dataset, it would be beneficial to manually verify the sampled queries against the images in the evaluation query set, thereby ensuring the inclusion of any potentially overlooked targets.

In addition to the aforementioned evaluation query sets, we created a challenging dataset to evaluate the system's performance in retrieving very small objects in the image. This evaluation query set contains 1,000 queries and 1,000 images. To obtain the captions corresponding to small objects, we applied a filter based on the area of the bounding box, selecting captions where the area is less than 5% of the area of the whole image. After filtering, we followed a similar procedure as described above to create the challenging evaluation query set.

---

[2] https://huggingface.co/sentence-transformers/all-MiniLM-L6-v2

Furthermore, we aimed to explore the system's ability to retrieve compositional queries. For this purpose, we created a compositional query set, where each query is obtained by sampling two random captions from an image and concatenating them together with a semicolon. The compositional query set also includes 1,000 queries and 1,000 images.

Lastly, inspired by the zero-shot performance of the CLIP model on ImageNet classification, we were interested in evaluating our system's classification performance. We included the ImageNet validation dataset, consisting of 50,000 images and 1,000 classes, to assess the system's ability to capture general information in whole images, in addition to the fine-grained details measured in the previous evaluations.

## C  QUANTITATIVE METRICS ON EVALUATION QUERY SET OF DIFFERENT INDEX SIZES

To showcase the performance of our ColCLIP Large model across various index sizes, we evaluated it on the simple evaluation query set with index sizes of 100, 1000, 5000, and 10808. Additionally, we included the performance of the CLIP Large Finetuned model for comparison. The results of these evaluations are presented in Table 7.

| Index Size | Model | AvgPrecision | Success@K | | | Precision@K | | | Recall@K | | |
|---|---|---|---|---|---|---|---|---|---|---|---|
| | | | K=1 | K=5 | K=10 | K=1 | K=5 | K=10 | K=1 | K=5 | K=10 |
| 100 | CLIP Large Fine-tuned | 0.680 | 0.546 | 0.857 | 0.929 | 0.557 | 0.180 | 0.099 | 0.542 | 0.855 | 0.927 |
| | ColCLIP Large | 0.704 | 0.569 | 0.864 | 0.935 | 0.583 | 0.181 | 0.099 | 0.569 | 0.859 | 0.930 |
| 1000 | CLIP Large Fine-tuned | 0.364 | 0.234 | 0.499 | 0.634 | 0.242 | 0.110 | 0.072 | 0.228 | 0.501 | 0.638 |
| | ColCLIP Large | 0.406 | 0.281 | 0.536 | 0.677 | 0.293 | 0.117 | 0.076 | 0.281 | 0.535 | 0.674 |
| 5000 | CLIP Large Fine-tuned | 0.194 | 0.116 | 0.266 | 0.375 | 0.136 | 0.069 | 0.051 | 0.113 | 0.262 | 0.366 |
| | ColCLIP Large | 0.211 | 0.126 | 0.286 | 0.383 | 0.149 | 0.075 | 0.054 | 0.126 | 0.286 | 0.381 |
| 10808 | CLIP Large Fine-tuned | 0.135 | 0.071 | 0.206 | 0.266 | 0.092 | 0.057 | 0.039 | 0.071 | 0.200 | 0.258 |
| | ColCLIP Large | 0.151 | 0.101 | 0.210 | 0.291 | 0.121 | 0.058 | 0.042 | 0.094 | 0.198 | 0.275 |

Table 7: Quantitative Metrics for Retrieval Models on Simple Query Set of Various Index Size

## D  ADDITIONAL EXPERIMENT WITH NUMBER OF SAMPLED CAPTION PER IMAGE (CPI)

As described in Section 3.2, during the training of the ColCLIP model, we chose to sample 5 captions per image (cpi = 5) at each training step. In additional experiments, we found that this strategy resulted in slightly better performance compared to the alternative strategy of sampling 1 caption per image (cpi = 1) as shown in Table 8. The validation loss curve also demonstrated faster convergence for the cpi = 5 strategy as shown in Figure 5.

| Index Size | CPI | AvgPrecision | Success@K | | | Precision@K | | | Recall@K | | |
|---|---|---|---|---|---|---|---|---|---|---|---|
| | | | K=1 | K=5 | K=10 | K=1 | K=5 | K=10 | K=1 | K=5 | K=10 |
| 100 | 1 | 0.695 | 0.567 | 0.854 | 0.936 | 0.580 | 0.179 | 0.099 | 0.562 | 0.852 | 0.933 |
| | 5 | 0.704 | 0.569 | 0.864 | 0.935 | 0.583 | 0.181 | 0.099 | 0.569 | 0.859 | 0.930 |
| 1000 | 1 | 0.382 | 0.249 | 0.521 | 0.661 | 0.268 | 0.115 | 0.075 | 0.250 | 0.520 | 0.660 |
| | 5 | 0.406 | 0.281 | 0.536 | 0.677 | 0.293 | 0.117 | 0.076 | 0.281 | 0.535 | 0.674 |
| 5000 | 1 | 0.205 | 0.117 | 0.284 | 0.385 | 0.139 | 0.075 | 0.053 | 0.119 | 0.281 | 0.379 |
| | 5 | 0.211 | 0.126 | 0.286 | 0.383 | 0.149 | 0.075 | 0.054 | 0.126 | 0.286 | 0.381 |
| 10808 | 1 | 0.145 | 0.086 | 0.214 | 0.284 | 0.107 | 0.059 | 0.043 | 0.082 | 0.202 | 0.272 |
| | 5 | 0.151 | 0.101 | 0.210 | 0.291 | 0.121 | 0.058 | 0.042 | 0.094 | 0.198 | 0.275 |

Table 8: Quantitative Metrics for ColCLIP Large Trained on Different Sample Captions Per Image

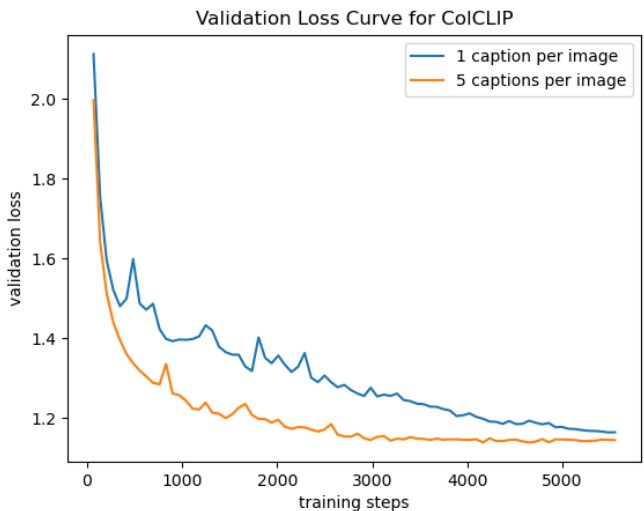

Figure 5: Validation Loss Curve For ColCLIP.

## E    LIMITATION OF SALIENCY MAP

As described in Section 4.7, it's essential to recognize the limitations of this saliency map technique. Firstly, when the object is substantially larger than the mask, the model can still make inferences about it, which makes the saliency map less effective. Additionally, if the query is associated with multiple similar objects situated in various locations within the image, the saliency map may not be effective. This is because the presence of any one object can result in a high alignment score, and masking any single object may not significantly impact the score.

## F    INTEGRATED GRADIENT ANALYSIS METHOD

As a baseline, we employ an empty image of the same size as the input image and construct a path from this baseline input to the actual input. Similarly, for the text inputs, we use the padding, represented by the end-of-sentence token in CLIP, with a length that matches the input query as the baseline. In our analysis, we leverage the logits of the MaxSim score between the image and the query for the Layer Integrated Gradient function.

Using Layer Integrated Gradient, we compute the gradients of the model's output with respect to the input at each interpolation point along the path. These gradients allow us to derive attribution scores for each word from the targeted text projection layer, indicating their importance in shaping the final text and image alignment results. Since our task involves a retrieval scenario with a single image and query, the target label always aligns with the desired label. Consequently, we have excluded this section from the visualization results and focused on the attribution scores and word importance.

