# OpenReview forum: "ColCLIP: Enhancing Fine-Grained Image Retrieval with Pre-trained Embeddings"
_ICLR.cc/2024/Conference — Submitted to ICLR 2024_

### Official Review · Reviewer_hnuQ · 2023-10-24

**Soundness:** 2 fair
**Presentation:** 2 fair
**Contribution:** 2 fair
**Rating:** 3
**Confidence:** 4

**Summary:**

This paper proposes a fine-grained image retrieval system that leverages and enhances the pre-trained embedding. The authors propose to fine-tune CLIP on the Visual Genome Dataset and incorporate the MaxSim operator for image-text interaction. The approach proposed by the authors outperforms the CLIP model on the Visual Genome dataset and the MSCOCO dataset.

**Strengths:**

1.	The proposed method surpasses the CLIP model on both the Visual Genome and MSCOCO datasets.
2.	This paper handles a potentially valuable practical scenario.

**Weaknesses:**

1.	This paper does not clearly define the research problem, its challenges, and how it differs from existing research directions. If it is focused on fine-grained retrieval in small areas of images, the authors should analyze why existing methods, even after fine-tuning, cannot achieve fine-grained retrieval.
2.	The novelty of this paper is limited. There have been various related studies on improvements to CLIP [1][2]. The authors should conduct a comprehensive comparative analysis of their proposed method with these approaches to emphasize the contribution of this paper.
3.	This paper lacks sufficient explanations for its design choices. The authors should explain why they chose to use the MaxSim operator instead of other designs [3][4]. Additionally, experimental comparisons should be conducted with the methods from [1][2].

[1] Li J, He X, Wei L, et al. Fine-grained semantically aligned vision-language pre-training[J]. Advances in neural information processing systems, 2022, 35: 7290-7303.

[2] Understanding and Constructing Latent Modality Structures in Multi-Modal Representation Learning

[3] Lee K H, Chen X, Hua G, et al. Stacked cross attention for image-text matching[C]//Proceedings of the European conference on computer vision (ECCV). 2018: 201-216.

[4] Zou X, Wu C, Cheng L, et al. TokenFlow: Rethinking Fine-grained Cross-modal Alignment in Vision-Language Retrieval[J]. arXiv preprint arXiv:2209.13822, 2022.

**Questions:**

Please refer to the "Weaknesses" section. Furthermore, it seems that the application scenario proposed by the authors could be valuable in the e-commerce context. Can the method proposed by the authors be adapted to datasets specific to this scenario?

---

### Official Review · Reviewer_coku · 2023-10-27

**Soundness:** 2 fair
**Presentation:** 3 good
**Contribution:** 1 poor
**Rating:** 3
**Confidence:** 4

**Summary:**

The paper proposes ColCLIP, a fine-grained image retrieval model based on the CLIP model which can enhance image and text embeddings for visual retrieval tasks. The authors integrated the CLIP model backbone and MaxSim operator during the model training for the downstream tasks. In the experiments, the author shows that ColCLIP outperforms better than standard CLIP in handling fine-grained retrieval tasks.

**Strengths:**

The paper is well-written and organized.
The training method and details are described clearly.
The experiments are conduct well

**Weaknesses:**

The key problem of this paper is: the fine-tuning of the CLIP model should be a popular topic after the release of the CLIP model and even before the CLIP model. The paper lacks background research and similar work comparisons, which makes it hard to tell how important this work is in the research community.

**Questions:**

what is the main innovation of the paper except the CLIP and MaxSum which is from other's work

---

### Official Review · Reviewer_yk6P · 2023-10-30

**Soundness:** 2 fair
**Presentation:** 1 poor
**Contribution:** 2 fair
**Rating:** 5
**Confidence:** 5

**Summary:**

This paper focuses on fine-grained image retrieval, which leverage pre-trained embeddings and enhance them specifically for fine-grained retrieval tasks. To address this issue, this paper employs a MaxSim operator to compute similarity for the interaction between image and text embeddings, which not only facilitates a more comprehensive level of interaction between images and queries
but also achieves precise alignment with the expected region. Extensive experiments show that the developed model could significantly improve the retrieval efficiency and accuracy. The subject matter addressed in this paper holds significant practical relevance, and the motivation behind it is evident. Regrettably, there are still certain issues pertaining to the writing style and experimental analysis that need to be addressed.

**Strengths:**

a.	This paper adapts MaxSim operator to the multimodal domain, which aims to utilizes the capabilities of a pre-trained multimodal embedding model with minimal modifications. Particularly, the method of this work could make original model acquire the capability to attend to diverse fine-grained details, irrespective of their prominence within the image.
b.	This paper utilizes saliency maps and integrated gradients to provide visual interpretations of the model's attention to different regions within the image and different segments of the text, which assists in obtaining a deeper understanding of the model's behavior.
c.	Extensive experiments are conducted to demonstrate the effectiveness and robustness of the proposed method.

**Weaknesses:**

a.	This paper should have a clear visual representation of the model's architecture. Without a model architecture diagram, it becomes challenging to assess the design choices of the proposed approach. The authors should include an illustrative model diagram in order to strengthen the paper's clarity and comprehensibility.
b.	This paper lacks sufficient explanation when describing the differences between the proposed method and MiniGPT-4. It should provide more detailed reasoning as to why MiniGPT-4 “is hindered by the costly imagetext interaction and the unpredictable nature of text output”.
c.	In Experiment part, this paper evaluates performance with few baselines. The proposed method whether helps the original different versions of CLIP to have a better performance is necessary to prove.
d.	This paper lacks a detailed illustration about the proposed method, for example, the image encoder in ColCLIP adopts the Vision Transformer from Krishna et al., but the relationship between original CLIP image encoder and the utilized image encoder should be explained.
e.	This paper needs to illustrate the related work more, it is advisable to illustrate that whether there exists some models which also focus fine-grained image retrieval by using CLIP.

Typos and minors
The whole manuscript should be carefully checked for typos, grammar and syntax, as there are many of them. The following are some example:
a.	The best value in Table 4, Table 5 and Table 6 should be highlighted.
b.	There is something different about the name of proposed model in Section 4.1 and Section 4.5.
c.	The formatting of the table could be enhanced to improve its aesthetic appeal.

**Questions:**

please see the weaknesses

---

### Official Review · Reviewer_Qpf8 · 2023-11-01

**Soundness:** 3 good
**Presentation:** 2 fair
**Contribution:** 2 fair
**Rating:** 5
**Confidence:** 4

**Summary:**

The paper introduces MaxSim layer into CLIP funetuning. MaxSim is an operation layer which constructs embeddings on top of all hidden states rather than a single token based on the original CLIP method. Combined with the targeting Visual Genome dataset, the authors hope their models can align between object specific queries to corresponding image regions. Leveraging the pretrained CLIP model weights, the paper showed improved text2image retrieval matching performance.

**Strengths:**

The framework fully leverages the benefits of pretrained CLIP models and shows improved matching performance for fine-grained text2image tests, to certain extent. The paper shows quantitative results for better active object regions on feature maps for their finetuned models. The approach also has reasonable reproducibility.

**Weaknesses:**

1) This work is mainly bringing in the main innovative layer of MaxSim from ColBERT to the proposed ColCLIP. It shows improvements on some setups but not all. For example, the ColCLIP base didn’t perform better than CLIP-based models, which left some concerns whether the gains come from only large model architectures or tied with specific datasets. 2) The paper discusses FILIP models and tries to differentiate from the proposed work. I am not convinced that FILIP was focusing on training from scratch and it is not a comparable approach. It indeed shares much with ColCLIP in particular the late interaction between text and image embeddings. It should be a strong baseline to this work. 3) At least one ablation study is missing in terms of the decision to remove symmetric contrastive loss. 4) The authors discussed retrieval quality for their Lite models but didn’t mention any latency or training time gains. It lost the opportunities to provide a comprehensive view of the motivation of dimensionality reduction.

Minor issues:
1. Proofread needed for spelling check, e.g. there are multiple places authors used "COLCLIP"
2. Image captions could be with bigger fonts for better visibility.

**Questions:**

It would be more sound if the authors can show results on other datasets, e.g. Flickr30K, or FILIP300M?

---

### Meta-Review · Area_Chair_xLhZ · 2023-12-05

**Metareview:**

No reviewers recommend acceptance; there are no author responses.

**Justification For Why Not Higher Score:**

no reviewers recommend acceptance

**Justification For Why Not Lower Score:**

n/a

---

### Decision · Program_Chairs · 2024-01-16

Reject